# Differentiation Syndrome in Acute Leukemia: APL and Beyond

**DOI:** 10.3390/cancers15194767

**Published:** 2023-09-28

**Authors:** Ashley C. Woods, Kelly J. Norsworthy

**Affiliations:** Center for Drug Evaluation and Research, U.S. Food and Drug Administration, Silver Spring, MD 20903, USA

**Keywords:** APL, AML, differentiation syndrome, ATRA, ATO

## Abstract

**Simple Summary:**

Differentiation syndrome (DS) is a frequent clinical complication of treatment with all-trans retinoic acid (ATRA) and/or arsenic trioxide (ATO), and is typically characterized by non-infectious related fever, dyspnea, hypotension, weight gain > 5 kg, pleural or pericardial effusions, and acute renal failure. DS was initially observed in patients with acute promyelocytic leukemia (APL) treated with ATRA and ATO but was later recognized with other targeted therapies for acute myeloid leukemia (AML). In this review, we discuss the pathogenesis, clinical manifestations, radiological findings, diagnosis, and treatment of DS in patients with APL treated with ATRA and ATO, as well as discuss DS in patients with AML treated with novel therapeutics.

**Abstract:**

Differentiation syndrome (DS) is a frequent and potentially life-threatening clinical syndrome first recognized with the advent of targeted therapeutics for acute promyelocytic leukemia (APL). DS was subsequently observed more broadly with targeted therapeutics for acute myeloid leukemia (AML). DS is typically characterized by fever, dyspnea, hypotension, weight gain, pleural or pericardial effusions, and acute renal failure. The incidence in patients with APL ranges from 2 to 37%, with the wide variation likely attributed to different diagnostic criteria, use of prophylactic treatment, and different treatment regimens. Treatment with corticosteroids +/- cytoreductive therapy should commence as soon as DS is suspected to reduce DS-related morbidity and mortality. The targeted anti-leukemic therapy should be discontinued in patients with severe DS. Here, we discuss the pathogenesis of DS, clinical presentations, diagnostic criteria, management strategies, and implementation of prospective tracking on clinical trials.

## 1. Introduction

Acute promyelocytic leukemia (APL) is a distinct sub-type of acute leukemia defined by the presence of the 15;17 translocation, resulting in the formation of the PML-RARA fusion protein, which blocks differentiation at the promyelocyte stage [1,2]. APL accounts for approximately 10–15% of new cases of acute leukemia [3]. Previously, APL was considered a fatal disease, with early deaths attributed to bleeding complications from disseminated intravascular coagulation (DIC) [4]. However, with the introduction of the differentiation therapies, all-trans retinoic acid (ATRA), and arsenic trioxide (ATO), and better supportive care to correct coagulopathy, the complete remission (CR) rates now often exceed 90% [5,6,7,8,9,10].

A common complication of treatment with ATRA and ATO is differentiation syndrome (DS), previously known as retinoic acid syndrome. While differentiation of the leukemic blasts is the desired objective with these targeted therapeutics, DS emerged as an important unintended consequence. DS was first described with the use of ATRA by Frankel et al. in 1992, with patients experiencing fever, respiratory distress, weight gain, edema, pleural or pericardial effusions, and episodic hypotension [11]. Since the first description of the syndrome in 1992, several studies have cited the incidence of ATRA and ATO-associated DS ranging from 2.5−37% [10,11,12,13,14,15,16,17,18]. The differences in incidences can be attributed to inconsistent diagnostic criteria, use of prophylactic corticosteroids in high-risk patients, and the use of gemtuzumab ozogamicin (GO) and/or chemotherapy induction regimens with ATRA and ATO [19]. Treatment with corticosteroids should be immediately commenced once DS is suspected to prevent DS-related morbidity and mortality. 

With the recent advancement of targeted therapeutics for treatment of acute myeloid leukemia (AML), DS is now frequently seen in this patient population as well. DS has been reported with the isocitrate dehydrogenase (IDH-1/2), FMS-like tyrosine kinase 3 (FLT3), and menin inhibitors [20]. The signs and symptoms of DS in AML appear mostly consistent with those seen in APL; however, there are some differences in clinical presentation. Additionally, unlike DS seen in APL, DS in AML can occur much later into treatment [20]. The emerging nature of DS as an adverse reaction in non-APL AML, and some differences in clinical presentation have led to missed cases with the IDH inhibitors ivosidenib and enasidenib, and an FDA Drug Safety Communication was released in November 2018 for enasidenib following fatal cases in the postmarketing setting [21,22]. Thus, it is critical to be aware of this toxicity to (1) recognize and promptly treat cases in clinical practice, and (2) implement close monitoring and management guidelines on clinical trials of novel targeted therapies for AML. 

In this review, we will discuss the pathogenesis, diagnostic criteria, prophylactic and treatment strategies for DS in APL, and provide an overview of novel therapeutic agents that cause DS in AML.

## 2. Differentiation Syndrome in APL

### 2.1. Pathogenesis

Although the exact pathogenesis of DS is not well understood, the hypothesis is that DS behaves like systemic inflammatory response system (SIRS) and capillary leak syndrome [19,23]. ATRA works in patients with APL by targeting the retinoic acid (RA) receptor and inducing terminal differentiation of APL blasts [23]. Differentiation of APL cells induces chemokine production in the lung and ATRA stimulation of APL cells increases migration towards alveolar epithelial tissues [24,25]. These chemokines serve as a chemoattractant for other inflammatory cells further exacerbating a hyperinflammatory state [26].

In addition, ATRA alters the adhesion properties of APL blast cells by inducing the expression of Beta-2 integrins, allowing for increased adhesions of APL blast cells to each other, the endothelium, and the sub-endothelial matrix [27,28,29,30]. The hypothesis of chemokine production and infiltration of alveolar and other organ tissues is supported by post-mortem studies of patients with APL and DS that showed diffuse neutrophilic infiltration with associated diffuse alveolar hemorrhage in the lungs and diffuse leukemic infiltration in various other organs [11,31,32].

### 2.2. Grading and Clinical Manifestations

The initial presentation of DS in patients with APL is non-specific and can mimic other etiologies such as infections or heart failure. The most frequent clinical signs or symptoms of DS are dyspnea, pulmonary infiltrates on chest radiograph, unexplained fever, effusions (pleural or pericardial), acute renal insufficiency, hypotension, weight gain (>5 kg), and edema [11,13,15,17]. As there is no single symptom or clinical manifestation of DS, the diagnosis is often made on exclusion of other possible etiologies and rapid clinical improvement upon initiation of intravenous corticosteroids [33]. The current standard of grading DS was proposed based on data from the PETHEMA study, wherein patients that have three or less signs or symptoms are considered to have moderate DS, while those with four or more symptoms are considered to have severe DS [15]. Table 1 lists the most common symptoms of DS.

In addition to the more commonly seen symptoms of DS in patients with APL, there have been several atypical presentations reported in the literature, including atypical cardiac manifestations, pancreatitis, and ocular manifestations [32,34,35,36,37,38,39,40]. The atypical cardiac presentations of DS include myocarditis, pericarditis, or myopericarditis [34,40,41,42]. These cardiac manifestations usually present with symptoms of chest pain and demonstrate EKG changes consistent with the cardiac pathology, such as QRS amplitude depression or diffuse ST elevations. In addition, elevations in troponin and pro-B-type natriuretic peptide (BNP) are usually present with these cardiac manifestations of DS. The more classical symptoms of DS usually co-occur with these atypical cardiac presentations; however, there was one case of isolated myocarditis as the only clinical symptom [34].

Ocular manifestations of DS typically present with a bilateral decrease in visual acuity, with ophthalmic examinations revealing sub-retinal fluids, retinal hemorrhages and lesions, retinal detachment, and macular edema [36,37,38,43]. Typical symptoms of DS may or may not be present with these ocular manifestations. Ocular manifestations of DS are thought to arise due to increased choroidal and retinal hyperpermeability and localized leukemic infiltration [43].

Other reported presentations of DS include pancreatitis, myalgias, acute febrile neutrophilic dermatosis (AFND) or Sweet’s Syndrome, and hyperbilirubinemia [33,35]. Clinicians should be aware of these diverse atypical symptoms of DS, such that they may be recognized and promptly treated when they arise. However, the incidence of these manifestations is sufficiently low such that they are not considered part of the primary diagnostic criteria for DS at this time.

### 2.3. Timing and Laboratory Findings of DS

The median day of onset of DS ranges from 7 to 12 days (range 0–46 days) [11,13,17,33,44]. Severe DS tends to occur earlier at a median day of onset of 6 days, compared to 15 days for moderate DS [15]. There is a bimodal distribution of the timing of DS, with nearly 50% occurring in the first week of treatment with ATRA and 50% developing in the third week or later of ATRA treatment for both moderate and severe DS [15,33]. Notably, DS in APL does not occur once patients achieve complete remission.

Laboratory abnormalities that are frequently seen with DS include leukocytosis, elevated creatinine, and elevated troponin and pro-BNP. Leukocytosis, defined as a white blood cell count (WBC) > 10 × 10^9^/L, can be accompanied by DS and can occur in nearly 50% of patients treated with ATRA + ATO [9]. However, ATRA can induce transient leukocytosis, which does not necessarily predict development of DS [45]. Nevertheless, studies have shown that a highly elevated WBC (>30 × 10^9^/L) is associated with development of DS and early mortality [45,46].

Acute kidney injury (AKI), presenting as elevated creatinine and blood urea nitrogen, is a listed criterion for DS and occurs in 11−66% of patients with DS [11,13,17,33]. Finally, in cases of hypervolemia or cardiac manifestations of DS, patients can develop elevated troponin and pro-BNP [19,34,40,41,42].

### 2.4. The Role of Imaging

Given the non-specific symptoms of DS, clinical correlation with radiographical findings is recommended. However, DS imaging can mimic other etiologies, and there are currently no radiographical findings pathognomonic for DS [47]. Pulmonary findings on chest radiographs are seen in up to 38% of patients with moderate DS and 80% in patients with severe DS [33]. Increased cardiothoracic ratio, increased vascular pedicle, pleural effusions, consolidations, and interstitial edema are common findings seen in chest radiographs in patients that develop DS [48,49,50,51]. Chest computed tomography (Chest CT) may be useful to further evaluate the lung parenchyma and to evaluate for the presence of pericardial effusions [47,48]. Davis et al. described the chest CT findings of patients diagnosed with DS and showed that small, peripheral nodules, pleural effusions, and ground-glass opacities were the most consistent chest CT findings [51]. Resolution of chest imaging findings may not exactly coincide with resolution of DS as the duration of complete resolution of pulmonary findings on imaging ranges from 3 to 42 days (mean, 11 days) [50].

There is a paucity of data on the use of preemptive imaging to predict DS. Karunakaran et al. performed daily chest ultrasounds on 35 patients with newly diagnosed APL on days 1–14 of ATRA + ATO therapy [52]. Three patients (8.5%) developed DS, with the “comet tail sign” (an indication of pulmonary edema) being detected 12 h prior to the onset of symptoms in one patient and at the time of initial presentation in the remaining two patients. At this time, preemptive chest imaging for monitoring of DS is not recommended, as up to 40% of patients with DS will have normal chest imaging [33].

### 2.5. Prevention, Treatment, and Management of DS

Use of corticosteroids for prevention of DS in patients with APL has been widely implemented, based on a few studies that demonstrated a reduction in DS-related complications in patients treated with prophylactic corticosteroids [13,53]. However, there has not been a prospective analysis on the benefit of corticosteroids in prevention of DS. Given the limited definitive supporting evidence for routine prophylaxis, there have been numerous preventative strategies evaluated utilizing different corticosteroids, including prednisone, dexamethasone, and methylprednisolone (Table 2) [6,9,19,54,55,56,57,58,59,60,61,62,63,64]. The duration of corticosteroid treatment for these preventative strategies varies ranging from a limited duration of therapy (5–15 days) to the entire duration of induction for patients treated with ATRA + ATO or ATRA + chemotherapy. In addition, some strategies gave corticosteroids to all patients, while some strategies only employed corticosteroids in patients that had a higher WBC (e.g., >5 × 10^9^/L). In the PETHEMA LPA99 trial, a 15-day duration of prophylactic prednisone was given to all patients, regardless of WBC [65]. In comparison to the PETHEMA LP96 trial, which only gave prophylactic dexamethasone to patients with a WBC > 5 × 10^9^/L, the incidence of severe DS was lower in the LPA99 trial (11.3% LPA99 vs. 16.6% LPA96, *p* = 0.07) [15,66]. However, this did not result in a significant decrease in DS-related mortality (1.1% vs. 1.4%) [66,67]. Studies have shown that a WBC >5–10 × 10^9^ /L, elevated creatinine, and presence of DIC are prognostic of development of DS [15,68]. Therefore, expert guidance has recommended use of prophylactic corticosteroids in patients who are at high risk of developing DS, defined as patients with WBC > 5 × 10^9^/L or elevated creatinine (>1.4 mg/dL) [19,66].

As mentioned previously, nearly half of patients treated with an ATRA-ATO based regimen will develop leukocytosis. The use of cytoreductive agents such as hydroxyurea, chemotherapy, or GO can be used to control hyperleukocytosis, as studies have shown that hyperleukocytosis is associated with development of DS and early death [45,46,66]. In extreme cases of hyperleukocytosis (typically defined as WBC > 100 × 10^9^), leukostasis can develop, which presents primarily as respiratory and neurological symptoms [69]. While the symptoms of leukostasis can be difficult to discern from those of DS (in particular, the pulmonary manifestations), the clinical management remains the same, with an emphasis on cytoreduction. However, the use of leukapheresis is generally not advised in patients with APL, due to the risk for precipitation of hemorrhagic events [45,70,71].

Once DS is suspected, treatment with dexamethasone intravenously at a dose of 10 mg every 12 h should immediately commence [33,70,71]. Treatment for suspected concurrent conditions, such as infection or congestive heart failure exacerbation, should occur simultaneously with treatment for DS [66]. If there is no improvement after twenty-four hours of treatment, dexamethasone can be increased to four times daily or every six hours, and should be continued until complete resolution of signs and symptoms [19,66,71]. Experts have recommended that ATRA and ATO should only be discontinued in cases of severe DS, severe renal or pulmonary dysfunction, or requirement of intensive care unit admission [19,66,71]. Once the DS symptoms resolve, ATRA and ATO can be restarted and continued until achievement of CR and/or completion of the specific APL treatment regimen [33].

Supportive measures, in conjunction with corticosteroid treatment, may be needed to effectively manage DS. Treatment for AKI typically consists of medical management with furosemide or other diuretic medications, but about 12% of patients have required dialysis for treatment of their AKI [72]. Furosemide can also be administered to treat peripheral and pulmonary edema and weight gain [19,33]. Mechanical ventilation (both invasive and non-invasive) may be required in patients with severe pulmonary failure [19,33,66].

## 3. Differentiation Syndrome with Novel Therapeutics for Acute Myeloid Leukemia

An overview of DS with novel targeted therapies for AML is listed in Table 3. 

### 3.1. IDH Inhibitors

Mutations in *IDH1* and *IDH2* occur in 6–16% and 8–19% of patients with AML, respectively [88]. The IDH1 and IDH2 inhibitors, ivosidenib and enasidenib, were initially approved by the United States Food and Drug Administration (FDA) for treatment of AML in 2018 and 2017, respectively. Mutations in IDH lead to a reduction in α-ketoglutarate and over-production of the oncometabolite 2-hydroxygluterate (2-HG), which leads to altered DNA methylation and impaired differentiation [21,73]. IDH inhibitors block 2-HG formation and induce myeloid differentiation in patients harboring *IDH1* or *IDH2* mutations [73,75,76]. This is believed to be mechanistically similar to the differentiation of APL blasts upon stimulation with ATRA [89]. FDA conducted a systematic analysis of DS cases from the ivosidenib and enasidenib marketing applications submitted to the Agency [21]. In the analysis, the rates of DS were 19% for both ivosidenib (34/179) and enasidenib (41/214). Most of the cases were moderate in severity, with eight severe cases seen with ivosidenib and five severe cases seen with enasidenib. There were two fatal cases of DS with each drug. On univariate analysis, patients with higher peripheral and bone marrow blasts and secondary AML had a higher relative risk of developing DS. However, on multivariate analysis, there were no patient characteristics that were significantly predictive of development of DS. The median time to the occurrence of DS with ivosidenib and enasidenib was 20 and 19 days, respectively; however, DS occurred as far out as 86 days into treatment. Treatment of DS primarily consisted of use of corticosteroids, with hydroxyurea and furosemide used less frequently. Due to the frequency of DS seen with ivosidenib and ensasidenib, both agents carry a boxed warning for DS in their USPI [14,74,77].

Another IDH1 inhibitor, olutasidenib, was recently FDA-approved for patients with R/R AML (*n* = 153) and a susceptible IDH1 mutation in 2022 [78]. On the pivotal trial (*n* = 153) that led to approval of olutasidenib, 25 patients (16%) experienced DS, with 14 patients (9%) experiencing ≥ Grade 3 DS [79,80]. The median time to onset of DS was 17.5 days, although one patient experienced DS on day 561 of treatment. Three patients permanently discontinued olutasidenib treatment due to DS and there was one fatal outcome attributed to DS [79]. As seen in patients who experienced DS with ivosidenib and enasidenib, most patients were treated with dexamethasone, hydroxyurea, and furosemide [79]. There is also a boxed warning for DS in the olutasidenib USPI [80].

Overall, the incidence of DS across the IDH inhibitor class has been reported in the range of 6–25% (Table 3). However, based on FDA’s systematic analysis, the adjudicated DS rates were the same at 19% for ivosidenib and enasidenib in patients with R/R AML and adjudicated rates for olutasidenib were 16% [21,80]. Therefore, the rates of DS are relatively consistent across the class.

### 3.2. FLT3 Inhibitors

FMS-like tyrosine kinase 3 (FLT3) activating mutations occur in up to 30% of patients with newly diagnosed AML and are associated with a poor prognosis [90]. FLT3 inhibitors induce terminal differentiation of leukemic blasts harboring FLT3 mutations [83,91]. There are currently three FDA-approved FLT3 inhibitors: midostaurin, gilteritinib, and quizartinib. Midostaurin was the first FDA-approved FLT3 inhibitor, achieving approval in 2017 for the treatment of adult patients with newly diagnosed AML harboring a FLT3 mutation in combination with standard induction chemotherapy [81]. In the pivotal trial that led to approval of midostaurin, there were no reported cases of DS occurring in patients using midostaurin and, to our knowledge, there have been no cases of DS attributed to midostaurin reported in the literature [82]. However, there have been case reports of AFND/Sweet’s Syndrome associated with the use of midostaurin in the literature that have resolved with corticosteroid treatment [92,93]. AFND/Sweet’s syndrome has been described with other FLT3 inhibitors previously as a manifestation of terminal differentiation of FLT3-mutated blasts [94,95,96].

Gilteritinib is a newer, selective, oral FLT3 inhibitor and was FDA-approved in 2018 for patients with relapsed/refractory AML harboring a FLT3 mutation as detected by an FDA-approved test [84]. In the pivotal trial that led to approval of gilteritinib, the FDA-adjudicated incidence of DS was 3% (11/319) [84,85], although lower rates have been reported in the literature [83]. The clinical manifestations of DS with gilteritinib were similar to DS seen in APL and with the IDH1 inhibitors. However, some cases of DS with gilteritinib had concomitant rash or AFND/Sweet’s Syndrome, similar to other FLT3 inhibitors. The onset of DS ranged from 2 to 75 days [84,97]. Of the 11 patients that experienced DS in the pivotal trial for gilteritinib, nine patients (82%) recovered after treatment of DS and there was one fatal case of DS. As with the IDH1 inhibitors, there is also a boxed warning for DS in the gilteritinib USPI [85].

Quizartinib is another oral, small molecule FLT3 inhibitor that is FDA-approved in combination with intensive induction chemotherapy and cytarabine consolidation, and maintenance following consolidation in adult patients with newly diagnosed AML that is FLT3-ITD positive [86]. Previously, quizartinib was investigated in a phase 2 and a phase 3 trials in patients with R/R AML harboring a FLT3-ITD mutation as monotherapy [98,99]. There were no reports of DS in the publications for these two large trials; however, the quizartinib USPI states that the rates of DS and AFND were 5% and 3%, respectively, in the R/R setting, an indication for which quizartinib is not approved [86]. In addition, there are two case reports of three patients with R/R AML that developed AFND/Sweet’s Syndrome after treatment with quizartinib [94,95]. These symptoms occurred 6–8 weeks after treatment with quizartinib and presented as erythematous nodules and eruptions. All three patients improved with corticosteroid therapy.

In summary, the incidence of DS across the FDA-approved FLT3 inhibitors has been reported in the range of 1–5% (Table 3), with FDA-adjudicated rates of 3–5% [85,86]. The lack of reported cases of DS with midostaurin may be related to its lack of single agent activity or the fact that it is administered in combination with cytotoxic chemotherapy; there were similarly no reported cases of DS observed with quizartinib in combination with chemotherapy [86,100]. However, the reported cases of Sweet’s syndrome with midostaurin indicate that this agent may have a similar differentiating effect and, thus, could in principal lead to development of DS. 

### 3.3. Menin Inhibitors

Menin inhibitors target the interaction of the KMT2A-menin inhibitor pathway, which inhibits leukemogenesis in patients harboring a KMT2A or NPM1 mutation [101]. KMT2A mutations occur in 5–15% of patients with acute leukemia, while NPM1 is more frequent occurring in nearly 30% of patients with acute leukemia [102,103]. Menin inhibitors disrupt the interaction between menin and MLL protein and induce differentiation of AML blasts harboring MLL or NPM1 mutations [104]. There are currently no FDA-approved menin inhibitors, although several are in development, including SNDX-5613 (NCT04065399), KO-539 (NCT04067336), JNJ-75276617 (NCT04811560), DSP-5336 (NCT04988555), DS-1594b (NCT04752163), and BMF-219 (NCT05153330), for treatment of acute leukemia.

SNDX-5613, or revumenib, was evaluated in 68 patients with R/R AML in a Phase 1 dose-escalation study in the United States [87]. DS occurred in 11 patients (16%), with all being reported as Grade 2 [87]. All cases of DS resolved with treatment with steroids and hydroxyurea, and one patient had their dose interrupted due to the event. There are currently no published clinical trial data on the incidence of DS with the other menin inhibitors in development. However, KO-539 was placed on partial clinical hold due to adverse events of differentiation syndrome, and there was a recent case series reporting fatalities due to DS in two patients that received treatment with an unknown menin inhibitor [105,106]. The partial clinical hold for KO-539 was removed once adequate safety monitoring was implemented for monitoring of DS [107]. 

### 3.4. Other targeted therapies

We note that other targeted therapies for the treatment of patients with AML have resulted in DS. The hypomethylating agents (HMA) azacitidine and decitabine, FDA-approved for the treatment of patients with MDS, are commonly used off-label for patients with AML, and have been reported to cause DS and AFND/Sweet’s syndrome [108,109,110,111]. Likewise, in the large, randomized, phase 3 trial (AGILE) of ivosidenib + azacitidine versus placebo + azacitidine in patients with newly diagnosed IDH1 mutated AML, the incidence of DS in the placebo arm was 8% (six patients) [74]. Although this rate was lower than what was seen in the ivosidenib + azacitidine arm (15%, 11 patients), these findings give credence to HMAs also causing DS. The 15% incidence of DS on the ivosidenib + azacitidine arm was reassuringgiven that combination of two agents with differentiating capabilities did not result in a higher incidence of DS than what was observed with ivosidenib alone. Furthermore, emerging targeted therapies, such as FHD-286 and ORY-1001, are being evaluated in patients with R/R acute leukemias and have been reported to cause DS. FHD-286, an inhibitor of BRG1 and BRM, is being evaluated as monotherapy in patients with R/R leukemia (NCT04891757) and was placed on full clinical hold in August 2022 due to fatal cases of DS [112]. The full clinical hold was removed once adequate safety monitoring and management plans were implemented in the protocol [113]. Finally, ORY-1001, or iadademstat, is a first in class lysine specific-histone demethylase 1A inhibitor that was evaluated in patients with R/R acute leukemia [114]. Out of 27 patients treated, two patients experienced DS, with one being fatal despite treatment with high-dose steroids.

## 4. Conclusions

DS is a well-known complication of treatment for APL with ATRA and ATO, but has increasingly been described in patients treated with targeted therapeutics for AML. With prompt initiation of steroids and supportive care, the mortality rate of DS is now only 1% in patients with APL treated with ATRA +/− ATO [66,67]. However, despite increased awareness of DS, it remains one of the leading causes of death for patients with APL [115]. There are still unanswered questions for prevention of DS in APL, such as whether all patients should receive prophylactic corticosteroids or only patients that are high-risk, and what the optimal duration of corticosteroid therapy should be.

Based on the observation of DS with targeted therapeutics first for APL, followed by several targets in AML, it is likely that any direct inhibition of a driver mutation in AML will lead to terminal differentiation and, hence, carry a risk of DS. Although there are similarities with DS seen in APL and DS seen in AML, features like AFND have emerged as more characteristic of certain targets (e.g., FLT3-mutated AML). Prophylactic corticosteroid therapy has not been evaluated for targeted AML therapies to date; this may be impractical for therapies that may cause DS several months into treatment. It also remains to be seen if the diagnostic criteria proposed by Montesinos et al. for APL are optimal for all targeted therapeutics for AML [15]. However, with the development of novel therapeutics for AML, it would be reasonable to employ diagnostic and therapeutic strategies learned from the APL and IDH-mutated AML experiences followed by an iterative approach that incorporates emerging clinical findings [15,21]. Investigational protocols should incorporate close monitoring and detailed management guidelines for DS based on current best practices to quickly diagnose and treat potential cases of DS with an aim to prevent serious morbidity and mortality. 

## Figures and Tables

**Table 1 cancers-15-04767-t001:** Reported frequency of clinical symptoms of DS in APL.

Symptom	Reported Rates in DS [11,13,15,17]
Dyspnea	84–100%
Unexplained Fever	74–100%
Pulmonary Infiltrates	52–100%
Weight Gain	50–100%
Effusions	36–100%
Hypotension	12–55%
Acute Kidney Injury	11–66%

**Table 2 cancers-15-04767-t002:** Prophylactic regimens used for prevention of differentiation syndrome.

	Induction Regimen	Prophylactic Corticosteroid	Dose	Duration	Patients Treated	Incidence of DS
Sanz et al., 1999 (LPA96) [54] *n* = 123	ATRA + Ida	Dexamethasone	10 mg BID	Not Reported	WBC > 5 × 10^9^/L	6% (7/123)
Sanz et al., 2004 (LPA99) [56] *n* = 426	ATRA + Ida	Prednisone	0.5 mg/kg daily	15 days	All patients	4% (18/426)
Powell et al., 2010 (C9710) [18] *n* = 481	ATRA + Dauno + AraC	None	N/A	N/A	N/A	37% (177/481)
Ravandi et al., 2009 [57] *n* = 82	ATRA + ATO ± GO	Methylprednisolone	20–50 mg daily	5–10 days	All patients	16% (13/82)
Sanz et al., 2010 (LPA2005) [55] *n* = 402	AIDA	Prednisone	0.5 mg/kg daily	15 days	WBC > 5 × 10^9^/L	29% (106/372 evaluable for DS)
Pei et al., 2012 [62] *n* = 73	ATRA + ATO ± HHT	None	N/A	N/A	N/A	5% (4/73)
LoCoco et al., 2013 (APL0406) [9] *n* = 162	ATRA + ATO vs. ATRA + Ida	Prednisone	0.5 mg/kg daily	Entire Induction	All patients	19% (ATRA + ATO; 15/77) 16% ATRA + Ida; 13/79)
Zhu et al., 2013 (APL07) [64] *n* = 231	ATRA + oral arsenic ± MTZ vs. ATRA + ATO ± MTZ	None	N/A	N/A	N/A	22% (51/231)
Burnett et al., 2015 (AML17) [6] *n* = 235	ATRA + ATO ± GO vs. ATRA + Ida	None	N/A	N/A	N/A	23% (55/235)
Iland et al., 2015 (APML4) [63] *n* = 124	ATRA + Ida + ATO	Prednisone	1 mg/kg daily	≥10 days	All patients	14% (17/124)
Zhang et al., 2018 (CCAPL2010) [60] *n* = 66 *	ATRA + ATO	None	N/A	N/A	N/A	14% (9/66)
Testi et al., 2018 (ICC-APL-01) [61] *n* = 258 *	ATRA + Ida	Dexamethasone	5 mg/m^2^	5 days	WBC > 10 × 10^9^/L	11%
Kutny et al., 2022 (AAML1331) [59] *n* = 154 *	ATRA + ATO ± Ida	Dexamethasone	2.5 mg/m^2^ BID	14 days	WBC > 10 × 10^9^/L	27% (41/154)

* = Pediatric patients. Abbreviations: AraC, cytarabine; ATO, arsenic trioxide; ATRA, all-trans retinoic acid; Dauno, daunorubicin; GO, gemtuzumab ozogamicin; HHT, homoharringtonine; Ida, idarubicin.

**Table 3 cancers-15-04767-t003:** Differentiation syndrome in novel therapeutics for AML.

Drug	Approved Indication(s) (Excerpted)	Reported Rates of DS	Median Day of Onset of DS (Range)
Ivosidenib [21,73,74]	ND-AML ≥ 75 years or comorbidities and R/R AML with IDH1 mutation	19–25%	20–29 (1–78)
Enasidenib [21,75,76,77]	R/R AML with IDH2 mutation	6–19%	19–30 (7–150)
Olutasidenib [78,79,80]	R/R AML with IDH1 mutation	14–16%	17.5 (1–561)
Midostaurin [81,82]	Combination with chemo in ND-AML with FLT3 mutation	NR	N/A
Gilteritinib [83,84,85]	R/R AML with FLT3 mutation	1–3%	NR (2–75)
Quizartinib [86]	Combination with chemo in ND-AML with FLT3-ITD mutation	5% (R/R setting)	NR
Revumenib [87]	Not approved	16%	18 (5–41)

Abbreviations: ND, newly diagnosed; NR = not reported; R/R, relapsed or refractory.

## Data Availability

Not applicable.

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
