# Peer review of "Differentiation Syndrome in Acute Leukemia: APL and Beyond"

_cancers, 2023, doi:10.3390/cancers15194767_

Round 1

Reviewer 1 Report

The article “Differentiation syndrome in acute leukemia: APL and beyond” is a well-written and informative article but there are some comments.

1. As differentiation syndrome (DS) is associated with leukocytosis, overlapping points of DS and leukostasis would be helpful to readers, these symptoms are hard to discern.

2. Mechanism of IDH inhibitors should be described more precisely. IDH1 and IDH2 are known to convert isocitrate to alphaketoglutarate (KG), and KG turns into 2HG. 

3. Signs and Symptoms of DS with incidence by a Table would be helpful to readers.

4. HMA are also noted for DS and comments on DS due to HMA or HMA combined with other agents might be more informative.

5. Comments or discussion on the difference in the DS incidence by IDH inhibitor and FLT3 inhibitor might be informative. Is there a possibility that reported DS by FLT3 inhibitor occurred by chance?

Reviewer 2 Report

The manuscript concerns  the well-known complication of APL treatment with ATRA and ATO – the differentiation syndrome (DS). In the first part, the authors present the history, pathogenesis, clinical manifestation, laboratory and  radiological examinations  and treatment of DS. These presented problems are well known and I propose to shorten them significantly.

In the next part, the authors  analyze the differentiation syndrome which can be observed  after novel therapeutics for AML. This part is correct and I have no comments on it.
